# Research Status of Manufacturing Technology of Tungsten Alloy Wire

**DOI:** 10.3390/mi14051030

**Published:** 2023-05-11

**Authors:** Jun Cao, Yongzhen Sun, Baoan Wu, Huiyi Tang, Yong Ding, Kexing Song, Chengqiang Cui

**Affiliations:** 1School of Mechanical and Power Engineering, Henan Polytechnic University, Jiaozuo 454000, China; syz@home.hpu.edu.cn; 2Chongqing Materials Research Institute Co., Ltd., Chongqing 400700, China; wubaoan@163.com (B.W.); hytang320@163.com (H.T.); 3Zhejiang Tony Electronic Co., Ltd., Huzhou 313000, China; dingyong1980@126.com; 4School of Mechatronics Engineering, Henan Academy of Sciences, Luoyang 471000, China; kxsong@haust.edu.cn; 5State Key Laboratory of Precision Electronic Manufacturing Technology and Equipment, Guangdong University of Technology, Guangzhou 510006, China; cqcui@gdut.edu.cn

**Keywords:** tungsten and tungsten alloy, manufacturing technology, research status, wire drawing process

## Abstract

In light of the fact that tungsten wire is gradually replacing high-carbon steel wire as a diamond cutting line, it is particularly important to study tungsten alloy wire with better strength and performance. According to this paper, in addition to various technological factors (powder preparation, press forming, sintering, rolling, rotary forging, annealing, wire drawing, etc.), the main factors affecting the properties of the tungsten alloy wire are the composition of the tungsten alloy, the shape and size of the powder, etc. Combined with the research results in recent years, this paper summarizes the effects of changing the composition of tungsten materials and improving the processing technology on the microstructure and mechanical properties of tungsten and its alloys and points out the development direction and trend of tungsten and its alloy wires in the future.

## 1. Introduction

The world’s tungsten resources are mainly concentrated in China, Russia, Vietnam and Spain, with China accounting for more than 50% of the world’s total reserves. The strategic position of tungsten resources is very prominent due to its global scarcity, its irreplaceability in industrial applications and its increasing importance in the national economy, national defense construction and high-tech industries. China has huge reserves of tungsten resources and has great potential for mining and development in tungsten manufacturing [1]. Tungsten is a rare metal with a high melting point, high density, high hardness, high wear resistance, high electrical conductivity and thermal conductivity, a high electron emission coefficient, a high compression modulus and elastic modulus, a low linear expansion coefficient, high-temperature creep resistance, magnetism resistance, corrosion resistance and a series of excellent and unique physical, mechanical and chemical stability properties. Its tungsten products—tungsten steel [2], cemented carbide [3], tungsten alloys, tungsten wires and tungsten compounds—are widely used in metallurgy, aviation, aerospace, ships, nuclear energy [4], machinery, electronics, automobiles, petrochemicals [5] and military industries and are known as “industrial teeth”, becoming an important functional structural material indispensable in industrial production. Figure 1 is the production process flow chart for pure tungsten wire.

In 1909, with the emergence of W. D. Coolidge’s ductile tungsten wire production process, tungsten wire began a long road of exploration as a lighting application [6]. Tungsten wires are classified into pure tungsten wires, doped tungsten wires and tungsten alloy wires, and tungsten wires are widely used as lamp filaments, heating elements, thermocouples and electron tube cathode materials due to their high temperature and high strength properties [7]. In recent years, with the increasing demand for diamond cutting lines in the photovoltaic industry, the original high-carbon steel wire has reached its limit, and the metal material needs to be further changed. Tungsten wire is gradually replacing high-carbon steel wire because of its high strength, good wear resistance and good machinability. Faced with the replacement problem of tungsten wire, the current tungsten wire needs to solve the problems of length, thinness and high strength. The author studied this problem and prepared tungsten alloy wire with a strength of at least 5500 MPa and a wire diameter of about 35 μm by controlling raw materials and processing, ensuring the length of the wire reached 10,000 m without interruption. Tungsten wires are used in argon arc welding electrodes, which can withstand higher currents and excellent performance from AZ80M magnesium alloy [8] and 443 stainless steel [9]. They have been prepared using a new additive manufacturing technique: wire arc additive manufacturing (WAAM). However, at low temperatures, the body-centered cubic tungsten metal shows greater brittleness and may fracture in daily applications, which largely limits the application of tungsten wires [10,11]. For this problem, a large number of studies have found ways to strengthen the properties of tungsten materials by adding various rare earth elements, oxides, rhenium, potassium, etc. [12,13,14]. By observing the microstructure, it was observed that the addition of these additions led to changes in the microstructure of tungsten materials, refinement of grains, etc., which improved their mechanical properties [15]. At present, the manufacturing processes of tungsten alloy wires are mainly powder manufacturing, pressing and sintering processes, rolling and spin-forging processes and wire drawing processes. Many studies have been performed by domestic and foreign scholars on how to improve the manufacturing process to improve the microstructure and mechanical properties of tungsten alloys.

## 2. Powder Manufacturing Technologies

The quality of tungsten powder directly affects the performance of tungsten and tungsten alloy products and plays a decisive role in the products. Therefore, the manufacturing and research of ultrafine nano tungsten powders are of vital importance in the tungsten industry. With the continuous improvement and development of modern technology, there are many new processes and methods in the manufacturing of tungsten alloys, such as hydrogen reduction, high-energy ball milling, cyclic redox, plasma technology, the sol–gel method, spray drying [16], etc. Patra A. et al. [17] prepared W–Mo and Y_2_O_3_ diffusion-reinforced W–Mo alloy powders by changing the composition of the metal powder using a high-energy ball mill. Mo alloy powders by changing the composition of the metal powders and found that the dispersion-reinforced tungsten alloys had a finer grain size and a higher density of sintered billets, which in turn exhibited better mechanical properties than the non-dispersion-reinforced tungsten alloys. Ammonium paratungstate (APT) is an important raw material in the production of tungsten in its industry; it is roasted to obtain WO_3_ (yellow tungsten oxide) and also reduced to obtain blue tungsten (blue tungsten oxide). Most tungsten and tungsten alloy powders are currently prepared from WO_3_ and blue tungsten. The blue tungsten powder contains a small amount of ammonia and has many surface pores, so it is highly active and easy to obtain fine tungsten powder during reduction. The current application of tungsten powder prepared from blue tungsten is becoming more and more widespread, and it is of great significance for the study of blue tungsten.

Xu Junliang et al. [18] selected APT with different particle sizes and loose packing densities as raw materials to prepare blue tungsten and studied the effects of calcination temperature, feeding rate and furnace tube speed (calcination time) on the physical properties (particle size, loose packing density, specific surface area, etc.) of blue tungsten. It was found that the higher the particle size and bulk density of APT, the higher the particle size and bulk density of blue tungsten obtained; the feed rate had less influence on the physical properties of blue tungsten; the higher the calcination temperature and longer the calcination time, the bulk density and specific surface area of blue tungsten gradually decreased. Wang Gang et al. [19] compared the differences between blue tungsten and purple tungsten hydrogen reduction methods for the manufacturing of tungsten powder under the same process conditions. The results showed that the tungsten powder produced by purple tungsten was finer in size and that the hydrogen flow rate and boat loading volume had less influence on the tungsten powder size. The effect of reduction temperature, hydrogen flow rate and reduction time on the morphology of tungsten powder was investigated by Yu Xiangbiao [20]. It was found that the reduction temperature had the greatest effect on the morphology of tungsten powder, followed by the hydrogen flow rate and reduction time. When the reduction temperature was 700 °C, the hydrogen flow rate was 150 mL/min and the reduction time was 55 min, the microstructure of tungsten powder appeared similar to that of purple tungsten, and the SEM image of blue tungsten microstructure is shown in Figure 2. Zhang Longhui et al. [21] studied the effect of using blue tungsten, purple tungsten and yellow tungsten as raw materials on the properties of the prepared tungsten powder and the strength of the pressed billets, and their results are shown in Table 1. The fisher sub-sieve sizer (Fsss) particle size of tungsten powder prepared by purple tungsten is 2.92 μm, the Fsss particle size of tungsten powder prepared by blue tungsten is 3.60 μm at maximum, and the Fsss particle size of tungsten powder prepared by yellow tungsten is 3.16 μm in between. The strength of the tungsten powder billets prepared from tungsten blue was 5.4 MPa, and the strength of the billets prepared from tungsten yellow was 4.4 MPa. The author studied the blue tungsten particle size and reduction parameters on the effect of tungsten powder particle size law and obtained the blue tungsten particle size of 35–42 μm, the reduction temperature of 800, 840, 900, 960 °C, hydrogen flow rate of 20 m^3^/h, the loading boat volume of 800 g conditions to obtain the tungsten powder of Fsss particle size of 2.0–2.2 μm without agglomeration, as shown in Figure 3.

Li Baoqiang et al. [22] used plasma spheroidization to prepare irregular tungsten powder into dense spherical tungsten particles, which have uniform pores and a high degree of densification, which is conducive to the manufacturing of tungsten-based composites with a tight structure. The thermal conductivity of the W–Cu composites prepared by it is about 10% higher than that of the W–Cu composites prepared from irregular tungsten powder, and the relative density also reaches 99.5%, indicating that spherical powder particles have certain advantages in the manufacturing of W–Cu composites. The aerosol method is to use high speed and high pressure airflow or other methods to turn the liquid metal into small droplets, such as mist, and then solidify them into powder particles. Because the powder prepared by this method has the advantages of a spherical structure and a small particle size, it is now one of the main methods for preparing metal powder. Yu Weirong et al. [23] used aerosolization to prepare CoCrMoW alloy powder, which was mainly spherical with some satellite particles on the surface. The average particle size of the powder was 37.5 μm, and its powder organization was mainly composed of dendritic and cytosolic crystals. Then SLM (selective laser melting) forming was carried out to obtain an alloy with a tensile strength of 1283 MPa, a yield strength of 852 MPa and an elongation of 7.9%.

Recently, the research team of the Russian Academy of Sciences prepared bimodal tungsten powder by the electric explosion wire method and then carburized tungsten powder. They obtained the best manufacturing process, which further improved the yield of tungsten carbide. Their results show that the optimum carburizing temperature of tungsten bimodal powder is 1200 °C, the heating time is 8 h and the molar ratio of C/W is 1.4. On this basis, the yield can reach 99% [24]. Tungsten powder was prepared by plasma rotary electrode processing (PREP). The rotary electrode was heated by a plasma gun to melt the cross section of the tungsten electrode bar to form a liquid film, which was thrown out by the centrifugal force of high-speed rotation and then atomized in an inert gas atmosphere to form a spherical powder. Li Xiaohui [25] and others prepared tungsten powder by plasma rotating electrode processing and analyzed its composition, morphology and properties. The micromorphology of tungsten powder is dendritic crystal and cellular crystal, as shown in Figure 4a. The particle size of tungsten powder is 15–53 μm, the surface is smooth and there is almost no hollow powder. As shown in Figure 4b, its particle size is 45–150 μm. The smoothness of powder particles of different sizes is different, which is related to the cooling rate. The smoothness of small-sized particles with a high cooling rate is slightly higher than that of large-sized particles with a low cooling rate. The results show that the powder prepared by PREP has a high spheroidization rate, uniform particle size distribution, high purity and a smooth surface. Yang Xingbo [26] and others prepared pure W and W–La powders by plasma rotary electrode processing and analyzed the influence of adding La_2_O_3_ on the particle size and morphology of the powders. It is found that the W–La powder obtained by adding La_2_O_3_ improves the yield of fine powder by reducing the surface tension of droplets, but at the same time increases the spheroidizing time of particles, resulting in some defects on the surface of the alloy powder.

Table 2 lists the different tungsten powder manufacturing methods in the literature. The size range of tungsten powder prepared by the hydrogen reduction method and the electric wire explosion method is wide. The tungsten powder prepared by plasma rotary electrode processing (PREP) has a high degree of spheroidization and a smoother surface. After adding La_2_O_3_, the yield of tungsten powder is improved, which provides an idea of how to improve the yield of tungsten powder. At present, powder preparation technology is devoted to preparing powders with higher sphericity, smaller scale and higher purity.

## 3. Powder Press Forming

The main methods of pressing and forming tungsten powder are moulding, cold isostatic pressing and hot isostatic pressing. The properties of the tungsten-pressed blanks prepared by different pressing methods are also very different. In conventional moulding, Shi Jun et al. [27] found that the density of pure tungsten billets and sintered bodies increased with increasing pressing pressure in the range of 100–400 MPa. The microstructure of pure tungsten prepared at different pressing pressures is shown in Figure 5. From the figure, it can be seen that the density of the grains keeps increasing with the increasing pressing force, the connection between grain boundaries is better, and the pores are reduced. When the pressing pressure is 400 MPa, the sintered body density reaches a maximum of 18.3 g/cm^3^, with a relative density of 94.8%. This forming method is limited by the tooling, and it is also only suitable for processing tungsten materials of small dimensions and simple structures. With the rapid development of tungsten and tungsten alloys in recent years, it is difficult to meet the growing demand for products prepared by this forming process. The isostatic pressing method prepares tungsten blanks with almost the same density and uniform distribution because the pressing process is subjected to uniform forces in all directions. Hot isostatic pressing is based on the development of cold isostatic pressing and is a method in which both temperature and pressure are applied to the tungsten powder. Hot isostatic pressing has received increasing attention due to its ability to obtain tungsten blanks with high densification and high performance, and more and more research has been carried out on this basis with excellent results [28]. Yi Zou et al. [29] used the multi-particle finite element method to numerically simulate the hot isostatic pressing process of pure tungsten powder to investigate the effects of pressure, temperature and particle size on the densification process of tungsten powder, which provides an idea for a more accurate understanding of the metal particle. Hu Biao et al. [30] used Marc to simulate the densification process of 93W–Ni–Fe during hot isostatic pressing, and the relative density of the tungsten alloy reached 93.68% at 1400 °C and 140 MPa pressure conditions. Experimental verification was also carried out, and the experimentally derived results were compared with the simulation results, which showed an average relative density error of only 1.32%. The experiments also revealed that the relative density increased by 2.3% at 1400 °C compared to 1300 °C under the same pressure conditions. Wang Yuanjun et al. [31] used the finite element analysis software DEFORM-2D to simulate the hot isostatic pressing process of W–Cu alloy to investigate the effects of temperature, pressure and holding time on the densification process of tungsten billets, and the optimum process parameters derived from the simulation were 950 °C/110 MPa/2 h. The average relative density obtained from the simulation was 0.3%, and the maximum error was 1.28% compared with that obtained from the experiment.

Wang Jianning et al. [32] prepared four kinds of 90W–Ni–Fe–Cu alloys with different Cu additions by hot isostatic pressing. The relative densities of all four tungsten alloys were above 98.5%, and the SEM images of tungsten alloys with different Cu contents are shown in Figure 6. With the gradual increase in Cu content, the tungsten content in their alloys gradually decreased, the higher the density of the microstructure, and the voids and pores gradually decreased, indicating that the addition of copper can inhibit the dissolution and precipitation of tungsten in the alloy phase. The tensile strength of the alloy reached a maximum of 953 MPa at 4% Cu addition. M. Eddahbi et al. [33] prepared two composites, W/CuY and W/CuCrZr, by the hot isostatic pressing technique and measured the relative densities of 98.1% and 98.8% for the two materials, indicating a high degree of densification. Byun Jong Min et al. [34] used a combination of spark plasma sintering and hot isostatic pressing to investigate the manufacturing process of dense tungsten, and after spark plasma sintering, the relative density of the sintered body reached a maximum of 91.7%. After hot isostatic pressing at 1650 °C for 2 h, the relative density reached a maximum of 97.2%, while the maximum bending strength increased to 761 MPa. Hu Biao and Cai Gaoshen [35] used the hot isostatic pressing process for 93 W–Ni–Fe sintered rods and powders, respectively, and obtained a relative density of 98.8% for the sintered rods and 92.7% for the powders when the temperature was increased by 100 °C, with relative densities increasing by 0.86% and 2.3%, respectively. It was also found that increasing both temperature and pressure increased the relative density of the tungsten alloy, with temperature having a greater effect. In addition, it was also found that increasing both temperature and pressure increased the strength, elongation and face shrinkage of the material by tensile testing. The same conclusion was also obtained by Lang Lihui [36] and others who performed hot isostatic pressing experiments under different processes.

Table 3 shows the relative densities of the samples obtained by different manufacturing methods. In summary, hot isostatic pressing technology can significantly improve the densification of tungsten alloys and improve their properties, but how to accurately determine the process parameters such as temperature and pressure is a key issue in the process of hot isostatic pressing. Currently, with the gradual improvement of the experimental research on hot isostatic pressing and the use of finite element simulation of the hot isostatic pressing densification process, it can be seen that hot isostatic pressing technology has a very broad prospect in the field of tungsten alloy.

## 4. Sintering Process

The conventional sintering process for tungsten or tungsten alloys is liquid-phase sintering, in which W grains tend to grow and cause collapse and deformation of the material due to the large difference in solid–liquid density, which leads to the sinking and aggregation of W grains under the effect of gravity, a phenomenon also found by Onur Dinçer et al. [37] in their experiments. Xu Huan et al. [38] used a W–Ni–Fe alloy with low tungsten content (mass fraction of 60–80%) prepared using a solid phase sintering process (1300 °C for 1 h) and found that its porosity increased with increasing tungsten content, causing a decrease in the tensile strength and elongation of the alloy. The effect of tungsten content on the tensile strength and elongation of the tungsten alloy is shown in Figure 7, where the alloy has a maximum tensile strength of 260 MPa and an elongation of 2.3% when the W content is 60%. In order to obtain better performance from high-density tungsten alloy materials, new sintering processes such as two-step sintering, spark plasma sintering (SPS), microwave sintering, activation sintering and selective laser sintering have emerged, which are of great interest to scholars at home and abroad [39].

Li Xingyu et al. [40] sintered tungsten powder obtained by the two-step sintering method at 1300–1450 °C for 1 h and then held at 1200–1350 °C for 10 h. The relative densities of the sintered billets obtained were above 98%, and the grain sizes were 0.70–0.95 μm. The relative densities of the sintered billets obtained by ordinary sintering at 1100–1600 °C also reached 98.5%, but the grain size is larger and the grains are porous. Subsequent tests also revealed that the two-step sintered specimens had finer and more uniform microstructure and better mechanical properties. Li Xingyu et al. [41] subsequently optimized the two-step sintering process to achieve a sintered tungsten blank with a relative density of 99% and produced a sintered specimen with a relative density of 99.3% and an average grain size of 290 nm. Que Zhongyou et al. [42] used two-step sintering, heating the W–10Re alloy to 1200 °C and then sintering it at 1150 °C for 10 h to obtain tungsten–rhenium alloy sinter with a relative density of 98.4% and a grain size of 260 nm. Compared with normal sintering, the two-step sintering resulted in finer grains, a more uniform microstructure distribution and higher mechanical properties.

For ordinary sintering, microwave sintering is also gaining attention in the tungsten industry because of its uniform heating, high efficiency, environmental protection and energy savings [43]. G. Prabhu et al. [44] used microwave sintering to sinter pure tungsten powder and high-energy ball mill-activated tungsten powder at 2073 K, respectively, and obtained sintered samples with relative densities of 85% and 93%, respectively, within 6–7 h for the whole process. Xu Lei et al. [45] used pressure-free microwave sintering and hot-pressure microwave sintering to prepare tungsten–copper alloys. Experiments showed that the tungsten–copper alloys prepared by hot-pressure microwave sintering had better densities and properties, and the alloy structure was more uniform. The relative density of the tungsten–copper alloy reached 98.65% at a sintering temperature of 1100 °C and a pressure of 40 MPa. Wang Langlang et al. [46] prepared a tungsten–copper alloy by microwave sintering with copper–plated tungsten powder and sintered it at 1100 °C for 1 h. The relative density was 97.3%. The microstructure of WCu20 at different temperatures is shown in Figure 8. At 1100 °C, the particle distribution is relatively uniform, and there are few pores. The author obtained W–1.0% La_2_O_3_ tungsten bars with a diameter of 27.0 mm and a length of 800 mm by isostatic pressing and sintered them by an intermediate frequency furnace with the sintering parameters (sintering temperature: 0–2300 °C and sintering time: 27 h) and hydrogen flow rate of 10 m^3^/h. The organizational structure of the sintered tungsten bars is shown in Figure 9. From the figure, it can be seen that the tungsten bar has a uniform structure after intermediate frequency sintering, and the La_2_O_3_ strengthening phase is uniformly distributed.

Jiří Matějíček et al. [47] used the spark plasma sintering technique to sinter tungsten powder and studied the composition, porosity, mechanical properties and thermal diffusivity of sintered tungsten from several factors, such as the sintering environment, sintering conditions and powder size, which are helpful for further optimization of the spark plasma sintering process. Zhang Jingang et al. [48] prepared W–Ni–Fe alloy by spark plasma sintering technology. The relative density of 93W–5.6Ni–1.4Fe alloy prepared at 1050 °C and 50 MPa pressure reached 98.12%, with a grain size of 0.871 μm, a hardness of 84.3 HRA and a bending strength of 987.2 MPa. Zhang Shuaihao et al. [49] prepared an ultrafine crystalline tungsten–nickel–iron alloy with a relative density of 98.6% and a grain size of 0.27 μm at 950 °C and 150 MPa pressure by spark plasma sintering, which also achieved a high hardness of 1079 HV. Figure 10 shows the EBSD images of tungsten alloys sintered at 950 °C and different pressures, from which it can be seen that the grain shapes are irregular polygons with relatively uniform size distribution and no obvious grain There is no obvious grain orientation. A. Muthuchamy et al. [50] and Lakshmi Prasad B.S. et al. [51] prepared tungsten alloys by adding yttria-stabilized zirconia (YSZ) and rhenium metal by spark plasma sintering, respectively, and found that the relative density and mechanical properties of the sintered samples were improved.

In conclusion, a lot of research shows that samples with high relative density and strength can be prepared by the sintering process at present. This process stage mainly solves the problem of sample density and strength and prepares the blank with the required strength and characteristics. Compared with traditional sintering after pressing, hot isostatic pressing and new sintering processes (two-step sintering, microwave sintering, spark plasma sintering, etc.) can not only obtain sintered blanks with better compactness and mechanical properties but also save time and improve efficiency. At present, the new sintering process is in the research stage, and hot isostatic pressing technology is still used for large-scale industrial production.

## 5. Rolling Processing

As tungsten is prone to uneven deformation during the spin-forging process, it is in an uneven state of stress and strain, resulting in defects inside and on the surface of the tungsten workpiece. Due to high-temperature exposure to air for a longer period of time, oxidation of tungsten metal may occur; therefore, rolling before the tungsten spin-forging process can effectively reduce alloy defects and the degree of oxidation, resulting in a higher-quality product. Rolling mills are divided into two-roller, three-roller and four-roller rolling mills according to the number of rolls; they can be divided into single-seat mills and multi-seat mills according to the production method, etc. Figure 11 shows a schematic diagram of the rolling process. Liu Guirong et al. [52] conducted rolling tests to analyze the effect of microstructure and deformation on the mechanical properties of tungsten alloys in different directions under different deformation amounts. During the rolling process, the microstructure of the tungsten alloy changed significantly with the increase in deformation amount, and the longitudinal tungsten particles were gradually pulled into fibrous form. Li Yuan et al. [53] prepared pure tungsten plates with different rolling deformations by pressing, sintering and warm rolling of fine tungsten powder to study the microstructure and weave evolution of pure tungsten plates under different rolling deformations. The microstructural evolution of pure tungsten plates at different rolling deformation amounts included three processes: grain breakage, recrystallization and gradual grain growth. With the increase in rolling deformation, the weave type of pure tungsten plate changes significantly. The Vickers hardness of tungsten plates at different rolling deformations is shown in Figure 12, with the maximum Vickers hardness of 492.5 HV at 60% thickness reduction.

Lv Yongqi et al. [54] established a finite element model of the pure tungsten hot rolling process based on the constitutive equation to simulate the rolling process at 1300–1600 °C and 10–40% deformation. The results show that the rolling deformation has a greater influence on the rolling process than the rolling temperature. In addition, the rolling process is prone to cracks due to uneven deformation, which can be avoided by selecting the appropriate rolling process and improving the high-temperature mechanical properties of the sheet. Zhang Xiaoxin et al. [55] prepared pure tungsten plates with about 80% total deformation by unidirectional rolling (UNR), cross rolling (CRR) and clock rolling (CLR), respectively, and studied the effects of different rolling methods on their macroscopic texture and mechanical properties. The relative density of the samples prepared by the three methods is the same, about 99.8%, and the microhardness is almost the same, about 440 HV. For the bending strength, the lowest value of the unidirectional rolling sample is about 758 MPa, and the cross-rolling and clock-rolling samples reach 1280 MPa and 1317 MPa, respectively. The volume fractions of {100}, {110} and {111} weave in the samples prepared by the three different methods are shown in Table 4, with all three samples having the least amount of {110} weave, followed by the unidirectional rolled sample having a minimum of 14.61% of {111} weave compared to the other two samples.

Wang Xingang et al. [56] prepared W–Cu (18.5% Cu) alloy plates by rolling a total deformation of 75% using a new encapsulated rolling process. The rolling deformation significantly affected the mechanical and electrical properties of the W–Cu alloy. The relative density and hardness of the W–Cu alloy increased with increasing roll deformation, reaching a maximum of 99.8% and 457 HV at 75% deformation. The electrical conductivity, thermal conductivity and thermal expansion coefficient decrease significantly with an increase in rolling deformation. To improve the thermal conductivity of the material, a 3-hour annealing heat treatment was carried out in a vacuum furnace at 1000 °C. The thermal conductivity after annealing and during rolling is shown in Figure 13. By comparing Figure 13a,b, it was found that the thermal conductivity did not change significantly after 3 h of annealing. After a subsequent extension of the time to 5 h, it was found that the thermal conductivity of the alloy sheets increased significantly with increasing annealing time. The effect of rolling processing on the mechanical properties, thermal conductivity and thermal shock damage resistance of particle-reinforced tungsten alloys was analyzed and discussed by Luo Laima et al. [57]. After the tungsten blanks were rolled and then spin-forged and drawn into tungsten wires, the internal organization of the tungsten rods and wires was more uniform and the surface quality was higher, which also further reduced the cost and improved the production efficiency.

To sum up, the rolling process before rotary forging can effectively improve the material structure, improve properties and reduce surface defects. Compared with unrolled products, the products produced by this method have good quality, less metal burning loss and high production efficiency, which have been developed rapidly in recent years.

## 6. Rotary Forging Process

The tungsten bars or rods obtained by powder metallurgy or smelting need to undergo plastic processing in order to carry out the drawing process, and the process used before the drawing process is the rotary forging process. Rotary forging is the process of repeatedly forging a billet using a high-speed rotating rotary forging tool, which gradually decreases the cross-section and increases the length and changes the microstructure of the material, which in turn affects the material’s properties [58]. Yang Yongbin [59] investigated the effects of rotary forging and heat treatment on the organization and properties of 93 W alloy and found that the tungsten alloy after rotary forging had a non-uniform tissue distribution and the axial grains were elongated, while the outer layer was found to be more deformed than the central part. After heat treatment, the tensile strength of the alloy decreases slightly, but the elongation increases and the plasticity of the material is greatly improved, which is conducive to the subsequent plastic processing. By comparing the 90W–7Ni–3Fe alloy after vacuum liquid phase sintering and rotary swaging with 15% deformation, N. Kaan Çalışkan et al. [60] found that the relative density of the tungsten alloy after rotary swaging increased, and the changes in elongation and tensile strength are shown in Table 5. From Table 5, it can be seen that the elongation of the tungsten alloy increased from 2.7% to 7.3%, and the tensile strength also increased from 615 MPa to 1105 MPa. Osama [61] studied the effect of 10–50% deformation on 0.03% Y_2_O_3_ W–Ni–Fe alloy by different degrees of rotary forging. The results show that with the increase in deformation, the hardness and tensile strength increase gradually, and the elongation decreases. When the deformation is 50%, the hardness and tensile strength are increased by about 39% and 45%, respectively, compared with the initial sample.

U. Ravi Kiran et al. [62] investigated the effect of cyclic heat treatment and rotary forging on the mechanical properties of W–Ni–Fe alloys containing small amounts of Co and Mo and found that with an increasing number of cycles, the solubility of tungsten in the matrix increased and the yield, tensile strengths and elongation increased. The tensile and yield strengths of the tungsten alloy after three cycles plus rotary forging increased to 1343 MPa and 1293 MPa, respectively, and the elongation decreased significantly from 29% to 8%. Lenka Kunčická and Adéla Macháčková et al. [63,64] processed the sintered W–Ni–Co alloy by rotary forging at 20 °C and 900 °C, using numerical simulations with finite element analysis software combined with experiments to study the effect of both methods on the tungsten alloy. The experiments showed that rotary forging processing could improve the strength of the material, while rotary forging had a large residual stress at 20 °C and a relatively uniform residual stress distribution at 900 °C. Lin Zehua et al. [65] observed SEM of 95W–3.5Ni–1.5Fe alloy by rotary forging with different deformations. The tungsten grains were elongated, and the adhesive phase was also elongated into long strips distributed between the tungsten grains. As can be seen in Figure 14, the microstructure leads to cleavage fracture of tungsten alloy with 40% deformation and tearing of bonding phase, while the fracture mode of undeformed tungsten alloy is mainly transgranular fracture. With increasing deformation, the tensile strength of the tungsten alloy increases and the elongation decreases.

Zhang Linhai et al. [66] prepared tungsten rods by powder metallurgy and rotary forging and investigated the effects of 30–70% deformation on the organization and room-temperature and high-temperature mechanical properties of tungsten rods. It was found that the grain deformation and distortion in the cross-section of tungsten rods gradually increased with the increase in the deformation amount of rotary forging, and the tungsten rods underwent deformation strengthening leading to grain refinement; the degree of fibrillation of the longitudinal grain organization increased continuously. With the increase in deformation, the density and hardness of the tungsten rod first increase rapidly, then slowly increase and level off. The relationship between deformation, tensile strength and elongation is shown in Figure 15. The room temperature tensile strength of tungsten rods gradually increases with the increase of deformation, and the elongation at room temperature is almost zero due to the brittleness of tungsten; the high-temperature tensile strength and elongation of tungsten rods gradually increase with the increase of deformation.

The tungsten rods or bars are drawn into wire after rotary forging, but the material is prone to uneven deformation during the rotary forging process, and in this uneven state of stress and deformation, it can cause internal defects to reduce the quality of the material and even cracks leading to fracture. To improve the drawability of materials, an annealing treatment is generally used [67]. The rotary forging process of tungsten is essential for the tungsten wire drawing process, so it is important to control the parameters of the rotary forging process to obtain high-quality forged material, which lays the foundation for the drawing process.

## 7. Annealing Process

For tungsten or tungsten alloy materials that have undergone large deformation plastic processing, the large degree of deformation leads to uneven internal organization of the material and an uneven stress distribution state, while the plasticity of the material decreases significantly, which is not conducive to further processing in subsequent processes. In order to adjust the internal grain structure and improve the processing properties of tungsten and tungsten alloy materials, annealing treatment of the materials is required. Yu Ming et al. [68] and Thomas Larsen et al. [69] conducted isothermal annealing experiments on pure tungsten plates with different rolling volumes at 1250–1350 °C and 1125–1250 °C, respectively, and found that tungsten recrystallized during the annealing process; the hardness of pure tungsten plates with different rolling deformation volumes gradually decreased as the annealing time increased. After the initial pure tungsten plate was rolled and processed, the work hardening was significant and was eliminated after recrystallization annealing, so that the material returned to the initial state and the hardness was basically equal at the end. Liu Guohui et al. [70] conducted annealing tests on rolled 95WNiFe alloy to study the effect of annealing temperature on the organization and properties of tungsten alloy at 800 to 1450 °C. The results showed that recrystallization of the tungsten alloy occurred at 1200 °C. The tensile strength and elongation of tungsten alloys in each state are shown in Table 6. The tensile strength decreased from 1215 MPa to 1050 MPa, and the elongation increased from 3% to 8% when the annealing temperature was increased from 800 °C to 1200 °C. Therefore, the annealing temperature range of 800 to 1100 °C can be selected in order to obtain tungsten alloys with better overall performance.

Xiang Zan et al. [71] prepared tungsten plates by powder metallurgy with the addition of yttrium oxide dispersion strengthening and then studied the recrystallization behavior of tungsten plates at 50% hot rolling deformation by isothermal annealing in the range of 1250 to 1350 °C. Simulations using the JMAK model were performed to predict the recrystallization behavior of the tungsten plate during plastic deformation, and the results were in good agreement with the experimental results. The addition of yttrium oxide retarded the recrystallization process of tungsten plates compared to pure tungsten. Zhang T et al. [72] prepared a W–1wt%Re–0.5wt%ZrC alloy and annealed the tungsten alloy at temperatures between 1000 and 1700 °C to investigate the synergistic effect of elemental Re and ZrC particles on the thermal stability of the tungsten alloy. The results showed that the addition of rare earth elements Re and ZrC particles significantly increased the recrystallization temperature of the tungsten alloy (1600 °C to 1700 °C). R. Liu et al. [73] prepared W–0.5ZrC alloy rods using high-temperature spin-forging processing to study the effects of different annealing temperatures on the microstructure and mechanical properties of the tungsten alloy. It was shown that during the spin-forging process, the alloy grain size distribution was not uniform because of the different degrees of expression in different parts of the bar. Figure 16 shows the effect of annealing temperature on the Vickers hardness of the tungsten alloy. As can be seen in Figure 16, the hardness is maintained between 440 HV and 450 HV when the annealing temperature is increased from 1000 °C to 1500 °C. When the annealing temperature is increased to 1800 °C, its hardness gradually decreases to about 400 HV, which estimates the recrystallization temperature of the wrought W–0.5ZrC alloy to be about 1500 °C.

Zhong Changchi [74] studied the effect of the high-frequency annealing process on the organizational properties and service life of pure tungsten resistance spot welding materials and found that the grain size and hardness of tungsten electrode materials could be adjusted. The number of grinding times could be significantly reduced, and the number of spot welding times could be increased by reasonably controlling the high-frequency annealing process, which in turn improved the service life of the materials. Wang Hongjian and Yan Yutao [75], through annealing heat treatment of tungsten-rhenium alloy wires, found that annealing could significantly reduce the surface cracks and improve the bending strength of the alloy wires. L. Tanure et al. [76] studied the microstructure distribution of pure and potassium-doped tungsten wires at different annealing temperatures. The longitudinal section of grain boundary diagrams of pure and doped tungsten wires at different annealing temperatures are shown in Figure 17, where both wire materials maintain elongated grains in the initial state. As shown in Figure 17a, with the increase in annealing temperature, the grains of pure tungsten wire gradually become coarser and the grain size changes significantly. As shown in Figure 17b, the doped tungsten wire throws maintain the elongated microstructure at 1600 °C. At 2100 °C, the grains changed, and some small grains were enclosed between the large grains.

The author studied the effect of different annealing temperatures and annealing times of a W–1.0% La_2_O_3_ tungsten bar on the grain size of the tungsten bar, and the grain size was 800–1000/cm^2^ under the conditions of heat treatment temperature 2300–2400 °C and annealing time 0.2–0.3 s, as shown in Figure 18, which is favorable for the later drawing process.

Above all, the large deformation process of the material leads to work hardening, and the plasticity is greatly reduced, which is not conducive to the subsequent processes. However, after annealing, the material recrystallizes, which effectively improves the internal crystal structure, decreases the strength and improves the plasticity.

## 8. Wire Drawing Process

After the tungsten rod or bar is processed into a tungsten rod with a diameter of about 3 mm by rotary forging, the coarse and fine tungsten wires of various sizes can be obtained by drawing processes in different processes. Due to the high plastic–brittle transition temperature of tungsten and poor processing performance at low temperatures, hot drawing is generally used, and the heating temperature decreases with the decrease in tungsten wire diameter [77]. The drawing equipment is shown in Figure 19. Figure 20 shows a diagram of the wire drawing process. Residual stresses are also generated during the drawing process of tungsten wire, and S.M. Weygand et al. [78] and Manel Rodríguez Ripoll [79] used finite element simulation of the drawing process to verify this behavior. During the drawing of tungsten wire, as the tungsten wire diameter decreases, the degree of deformation increases and the rate of process hardening increases accordingly, affecting the quality of tungsten wire production and subsequent drawing, requiring intermediate annealing of the tungsten wire. Manel Rodríguez Ripoll et al. [80] used a viscoplastic self-consistent model (VPSC) and a crystal plasticity finite element model (CPFEM) to simulate the microstructure and texture evolution during the drawing process. Both models predicted the strain deformation during metal drawing, and CPFEM also reproduced the curl structure of the tungsten wire cross-section, the “Van Gogh’s sky structure”. Both models predicted the texture better for moderate deformation and slightly differed for large deformation, but were qualitatively consistent. Na-na Qiu et al. [81] prepared two tungsten rhenium alloy wires with different crystal sizes and investigated the microstructure and tensile properties of coarse (C-WRe) and fine (F-WRe) alloy wires from room temperature to 800 °C. Figure 21 shows the SEM micrographs of the tungsten alloy wires. It can be seen from the figure that the two alloy wires have similar curl structures in the cross-section, but the fine W–Re alloy wire has a finer fiber structure in the longitudinal section. At room temperature, the tensile strengths of the coarse W–Re and fine W–Re alloy wires were 3.5 G Pa and 4.4 G Pa, respectively, with an area reduction of 40.8% and 73.8%, respectively. At 800 °C, tensile strengths of 1.8 G Pa and 3.8 G Pa were achieved with area reductions of 73.3% and 75%, respectively. It can be concluded that the fine W–Re alloy wires have higher tensile strength and better ductility at room temperature and 800 °C.

R. Michel et al. [82] performed tensile experiments on tungsten wires between 25 °C and 320 °C. The experiments showed that the tensile strength of tungsten wires gradually decreased with increasing temperature, and their sectional shrinkage increased from 20% to 55%. Wang Luyan et al. [83] prepared lanthanum tungsten alloy wires (φ1.00 mm–φ0.50 mm) using different drawing process parameters to investigate the effects of different processes on the organization and properties of lanthanum tungsten alloy wires. The controlled mold temperature of each process was 550 °C, and the wire heating temperature (reduced by about 25 °C per pass) was 1050 °C and 950 °C, with the specific parameters shown in Table 7. By testing the hardness and straightness of the tungsten alloy wires prepared by different processes, it was found that the hardness and straightness of processes 2 and 4 were better than those of processes 1 and 3, with the highest hardness of process 2 reaching about 750 HV and straightness reaching 90%. From Figure 22, it is obvious that the grain size of the tungsten alloy wire cross-section prepared by each process is different, and the grain size prepared by process 2 and process 4 is a little smaller, among which the grain size of the tungsten alloy wire prepared by process 2 is the smallest, with an average grain size of 0.0982 μm. Through comprehensive analysis, it can be concluded that the best manufacturing condition is process 2.

P. Zhao et al. [84] investigated pure tungsten wires after different heat treatments and showed that after annealing at 1273 K for 3 h, the grains were still fibrous elongated grains compared to untreated tungsten wires; only the grain size increased, indicating that recrystallization of tungsten wires occurred at 1273 K. When annealed at 1900 K for 30 min, the grains grew into an equiaxed crystalline form. However, these cannot be clearly expressed for the range in which complete recrystallization of specific tungsten wires occurs, so additional temperatures are needed to allow a more detailed understanding of the recrystallization process of tungsten wires. Vladica Nikolić [85] et al. studied the effect of annealing heat treatment on pure and potassium-doped tungsten wires in the range of 900 °C to 1600 °C. It was shown that pure tungsten wires completed recrystallization in the range of 1300–1500 °C, where the fibrous structure of the grains disappeared and coarser grains were obtained. The doped tungsten wires maintained an overall elongated grain structure in this temperature range, indicating that the recrystallization temperature of doped tungsten wires was higher than 1600 °C. Vladica Nikolić et al. [86] investigated the effect of annealing heat treatment temperatures of 1300 °C and 1600 °C on the fracture behavior of pure and potassium-doped tungsten wires. It was found that the fracture toughness of both pure and doped tungsten wires decreased as the annealing temperature increased, but the decrease was significant for pure tungsten wires and not for doped tungsten wires. The fracture mode of pure tungsten wire was a mixed brittle fracture consisting of destructive and intergranular fractures after annealing at 1300 °C for 1 h. The fracture mode of doped tungsten wire was mainly obvious cleavage fracture and knife necking. When annealed at 1600 °C for 1 h, the pure tungsten wire had a similar fracture to that at 1300 °C. The fracture characteristics of the doped tungsten wire were more pronounced as the cleavage area increased. The author studied the fracture morphology of W–1.0% La_2_O_3_ wire drawing and concluded that the agglomeration of La_2_O_3_ in the matrix is the main cause of wire breakage during drawing, as shown in Figure 23.

The quality of tungsten wire is not only affected by the drawing process parameters but also by the lubrication and surface oxidation during the drawing process, which have a significant impact on the quality of the drawn tungsten wire. The particle size and viscosity of graphite milk, which is often used as a lubricant for drawing tungsten wires, can affect the generation of cracks and wire breakage during the drawing process. Graphite milk is different for drawing coarse tungsten wire [87] and fine tungsten wire [88], and it needs to be selected appropriately according to the thickness of the tungsten wire and the drawing principle. Huang Can-Xin [89] analyzed and studied the oxidation behavior of tungsten wire and proposed effective measures to improve the surface quality of tungsten wire.

Generally speaking, drawing tungsten wires will have different drawing processes due to the different types and characteristics of tungsten wires, drawing equipment and compression ratio. At present, it is necessary to do a lot of research to determine various process parameters in the process of wire drawing and solve the problems that affect the surface quality of silk thread and phenomena such as wire breakage and splitting.

## 9. Conclusions and Perspectives

The quality of the powder plays an indispensable role in the quality of the final product. Tungsten powder with higher purity, fewer impurities and finer grains can be prepared by optimized hydrogen reduction methods and new powder preparation methods (plasma spheroidization method, gas atomization method, electric wire explosion method, plasma rotary electrode processing (PREP), etc.). Moreover, the blank for processing products can be obtained by pressing and sintering processes. Compared with traditional sintering after pressing, the hot isostatic pressing process can not only obtain a sintered blank with better compactness and mechanical properties but also reduce the cost and improve production efficiency.

Then, through large plastic deformation processing (rolling, rotary forging) and annealing treatment, finished products or semi-finished products with mechanical properties that meet the requirements can be obtained. The work hardening phenomenon occurs after rolling and rotary forging, and the performance of the material shows that the strength is improved and the plasticity is greatly reduced, so it is difficult to carry out the wire drawing process. However, after annealing, the material recrystallizes, the grains grow again, and the plasticity of the material is improved, which is convenient for the wire drawing process. At last, the use of suitable drawing process parameters, good lubrication and the avoidance of problems such as wire breakage and splitting can produce high-quality, high-strength wire material.

In this paper, the main factors affecting the properties of tungsten alloy wires are briefly introduced. For example, various process parameters, composition of tungsten alloy, shape and size of powder, etc. In this paper, the preparation technology of tungsten alloy wire is introduced in detail, such as powder preparation, press sintering, rolling and rotary forging, annealing and wire drawing.

Future development directions and trends for tungsten and tungsten alloy wires can be summarized as follows:(1)At present, the new powder preparation method is not mature enough and is at the research stage, with high operational difficulties and high costs that prevent mass production. Therefore, research in this area should be increased and put into practical production as soon as possible.(2)At present, it has become a trend to prepare blanks using hot isostatic pressing technology, combining pressing and sintering. A new sintering technology is in the research stage. For practical production, hot isostatic pressing technology is still used, and new sintering technology still needs a lot of research.(3)For the drawing process, it is imperative to reduce costs and strictly control the drawing process parameters (drawing temperature, drawing speed, compression rate, etc.) to produce high-quality, high-strength wire. There is little research in this field, so more research on the wire drawing process is needed in the future.(4)With the extension of tungsten and tungsten alloy wire in the field of diamond wire cutting, the length of the wire is much longer than that used in the past (incandescent lamps, etc.). In the future, the fine diameter and length of silk thread will be difficult problems to solve.

## Figures and Tables

**Figure 1 micromachines-14-01030-f001:**
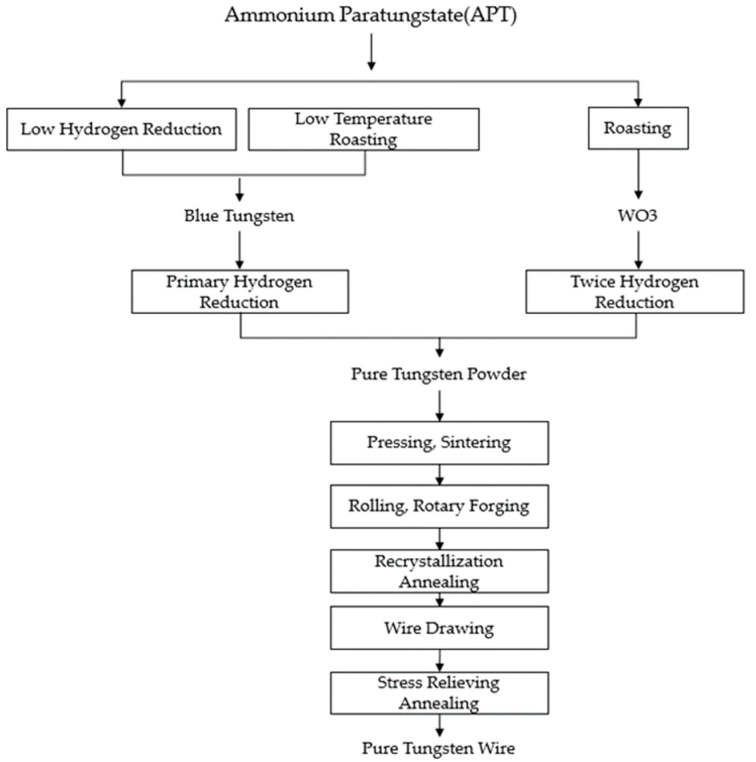
Production process of pure tungsten wire.

**Figure 2 micromachines-14-01030-f002:**
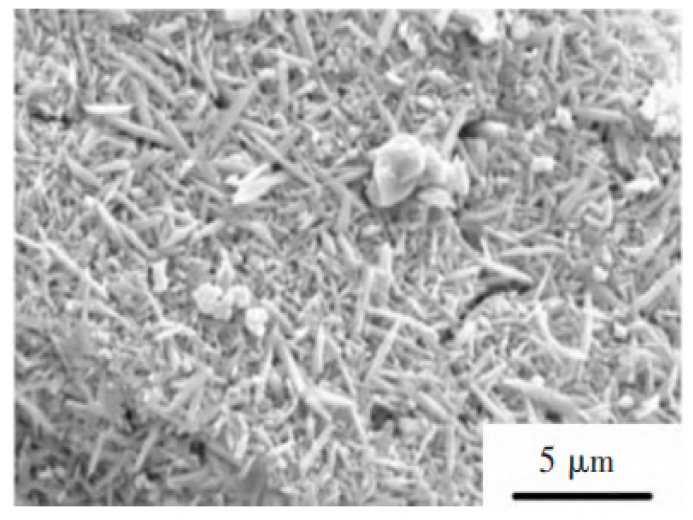
SEM image of the microscopic morphology of blue tungsten [20].

**Figure 3 micromachines-14-01030-f003:**
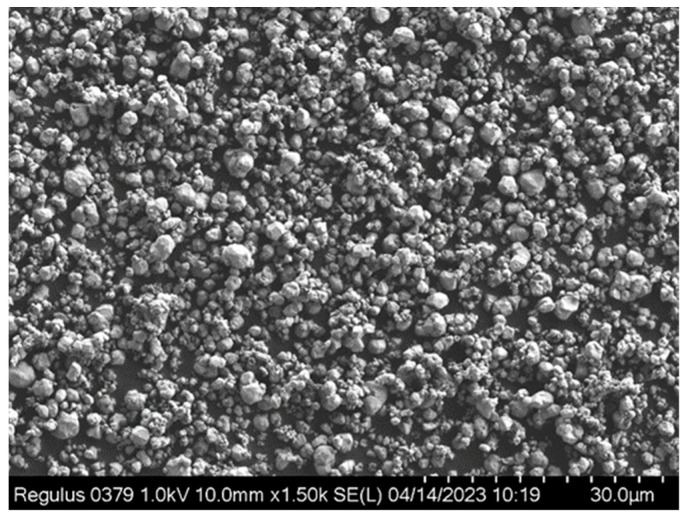
Tungsten powder with an Fsss particle size of 2.0–2.2 μm.

**Figure 4 micromachines-14-01030-f004:**
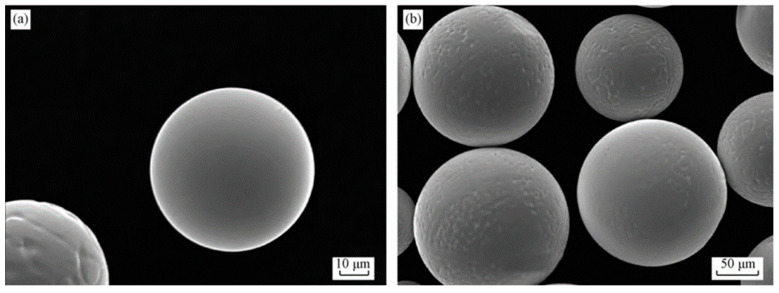
Surface morphology of tungsten powder [25]. (**a**) high cooling rate; (**b**) low cooling rate.

**Figure 5 micromachines-14-01030-f005:**
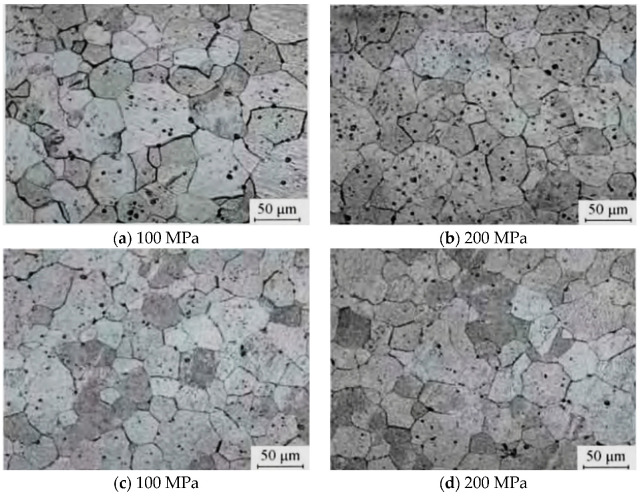
Microstructure of pure tungsten prepared by different pressing press pressures [27].

**Figure 6 micromachines-14-01030-f006:**
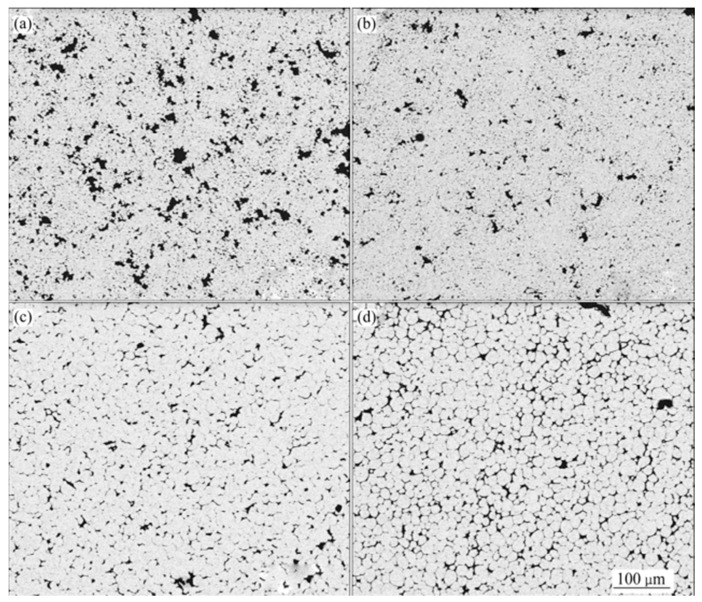
SEM images of tungsten alloys with different Cu contents [32]. (**a**) 0% Cu; (**b**) 2% Cu; (**c**) 4% Cu; (**d**) 6% Cu.

**Figure 7 micromachines-14-01030-f007:**
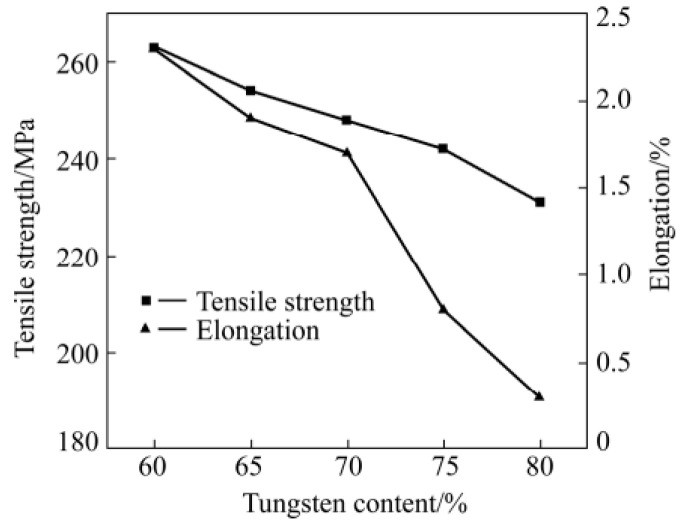
Effect of tungsten content on the tensile strength and elongation of tungsten alloys [38].

**Figure 8 micromachines-14-01030-f008:**
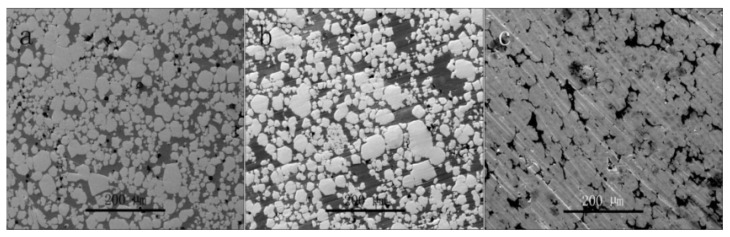
Microstructure of WCu20 at different temperatures [46]: (**a**) 1000 °C; (**b**) 1100 °C; (**c**) 1200 °C.

**Figure 9 micromachines-14-01030-f009:**
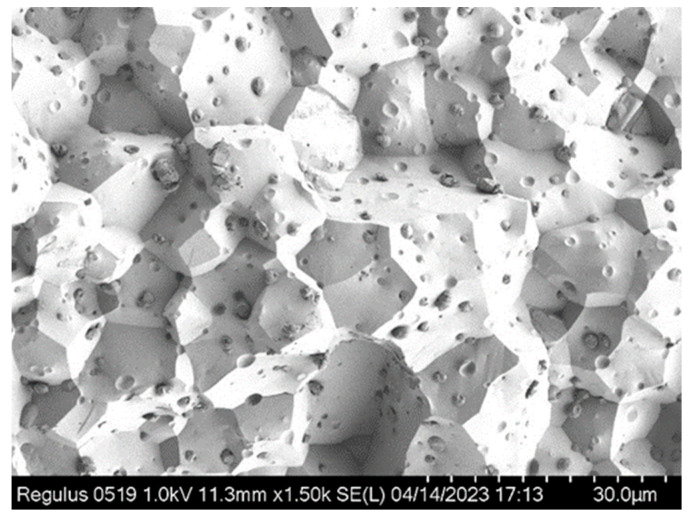
Fracture morphology of W–1.0% La_2_O_3_ after intermediate frequency sintering.

**Figure 10 micromachines-14-01030-f010:**
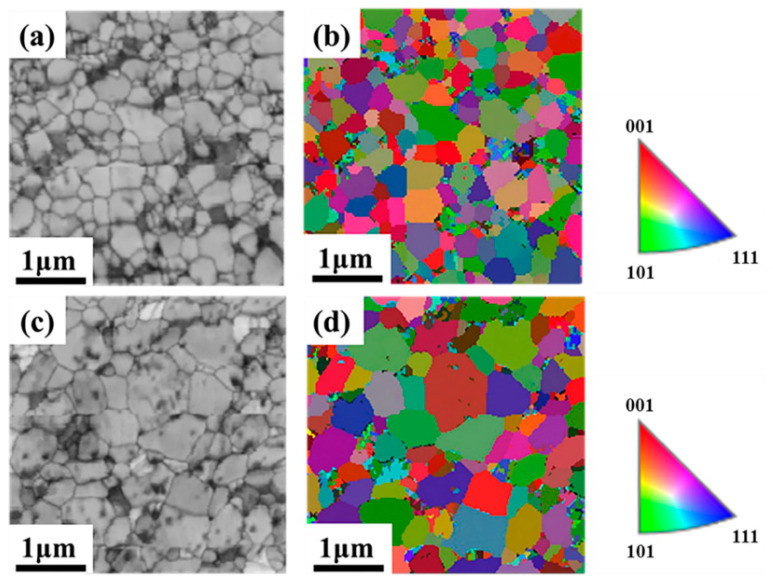
EBSD orientation diagram and inverse pole figure (IPF) of tungsten alloy sintered at 950 °C [49]: (**a**,**b**) 150 MPa; (**c**,**d**) 50 MPa.

**Figure 11 micromachines-14-01030-f011:**
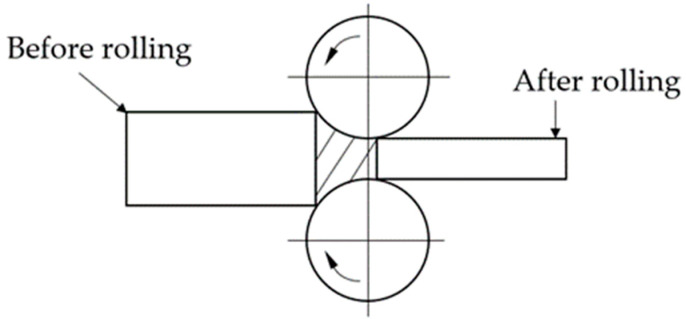
Schematic diagram of the rolling process.

**Figure 12 micromachines-14-01030-f012:**
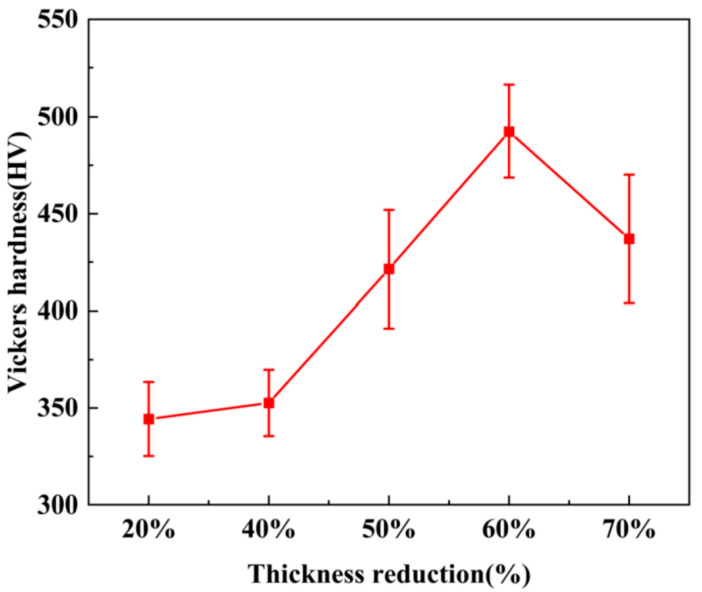
Effect of thickness reduction on Vickers hardness of tungsten plates [54].

**Figure 13 micromachines-14-01030-f013:**
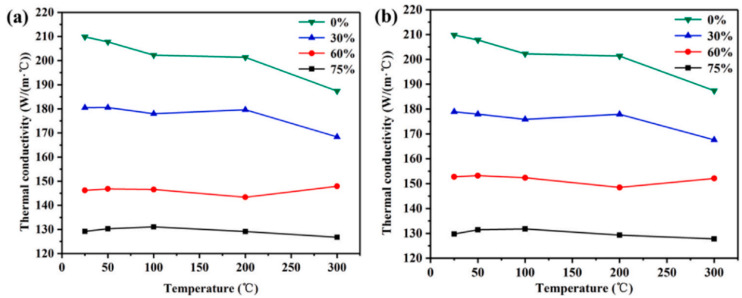
Effect of rolling deformation on thermal conductivity [56]: (**a**) rolling state and (**b**) annealed state (annealing: 1000 °C/3 h).

**Figure 14 micromachines-14-01030-f014:**
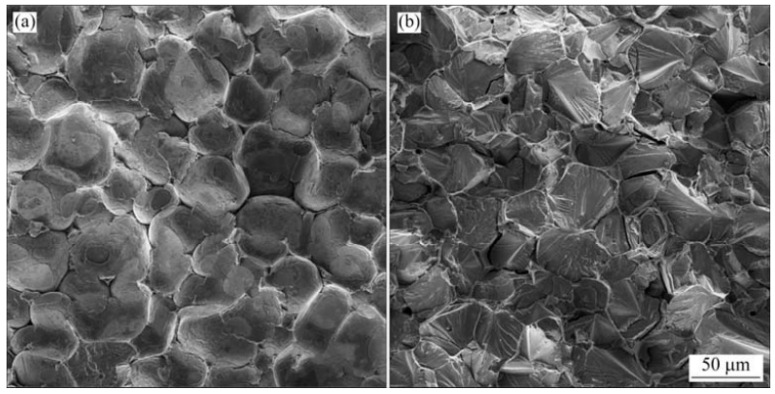
SEM morphology of tensile fracture of tungsten alloy [65]: (**a**) undeformed; (**b**) 40% deformation.

**Figure 15 micromachines-14-01030-f015:**
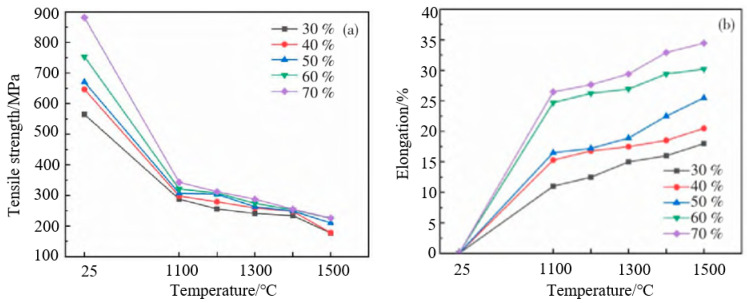
Relationship between the deformation of forged tungsten rods and tensile strength and elongation at room and high temperatures [66]: (**a**) tensile strength; (**b**) elongation.

**Figure 16 micromachines-14-01030-f016:**
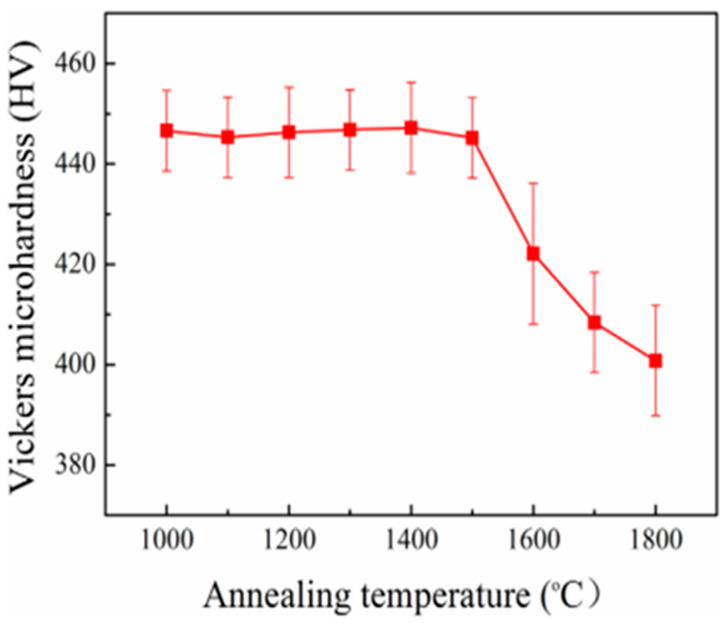
Effect of annealing temperature on Vickers microhardness of tungsten alloy [73].

**Figure 17 micromachines-14-01030-f017:**
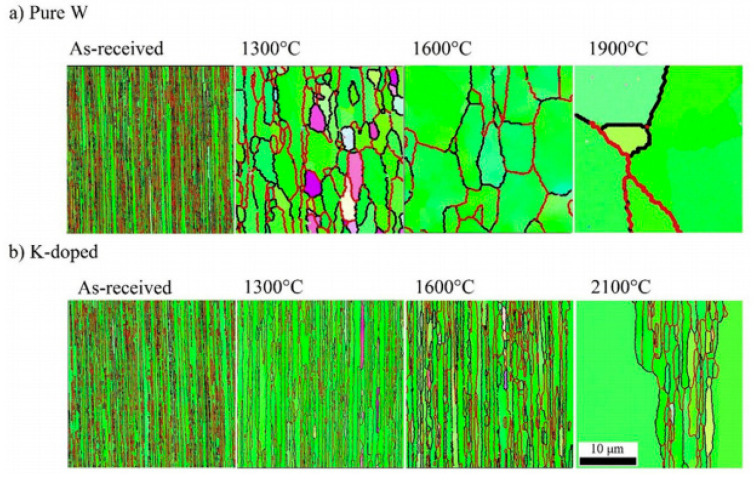
Grain boundary diagram of a longitudinal section of tungsten wire at different annealing temperatures [76]: (**a**) pure tungsten wire; (**b**) K-doped tungsten wire.

**Figure 18 micromachines-14-01030-f018:**
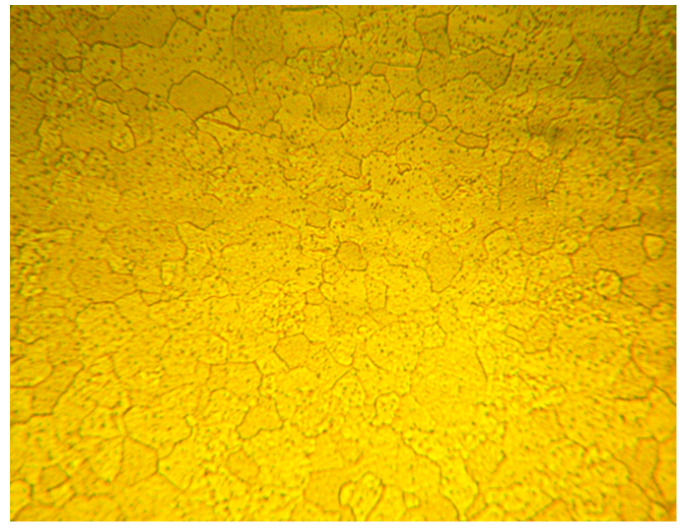
W–1.0% La_2_O_3_ tungsten bar’s annealed grains.

**Figure 19 micromachines-14-01030-f019:**
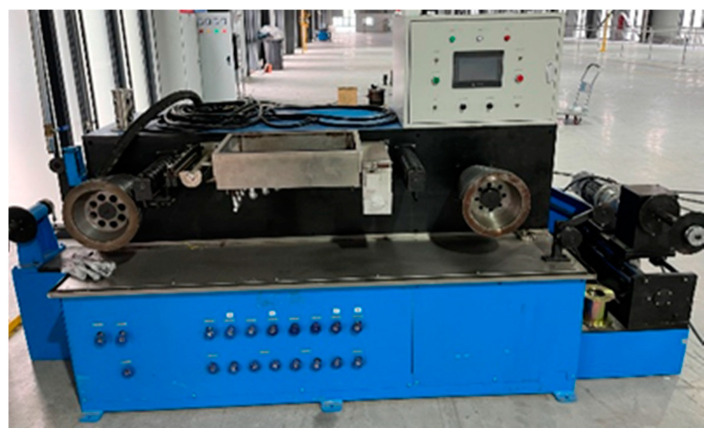
Tungsten wire drawing equipment.

**Figure 20 micromachines-14-01030-f020:**
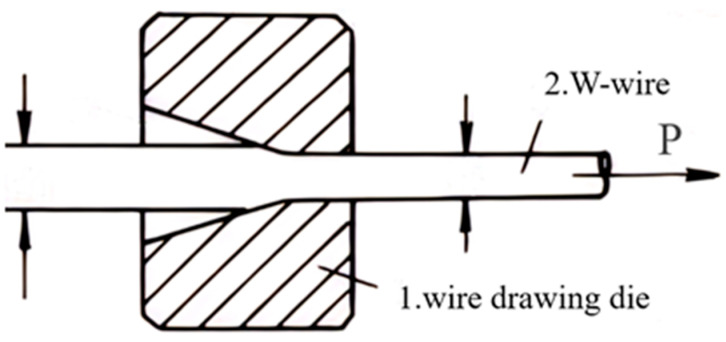
Schematic diagram of wire drawing processing.

**Figure 21 micromachines-14-01030-f021:**
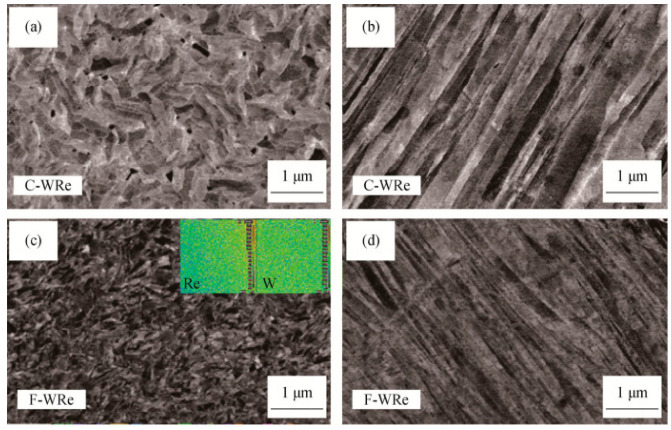
SEM images of tungsten alloy wire cross-sections [81]: (**a**,**c**) cross-section; (**b**,**d**) longitudinal section.

**Figure 22 micromachines-14-01030-f022:**
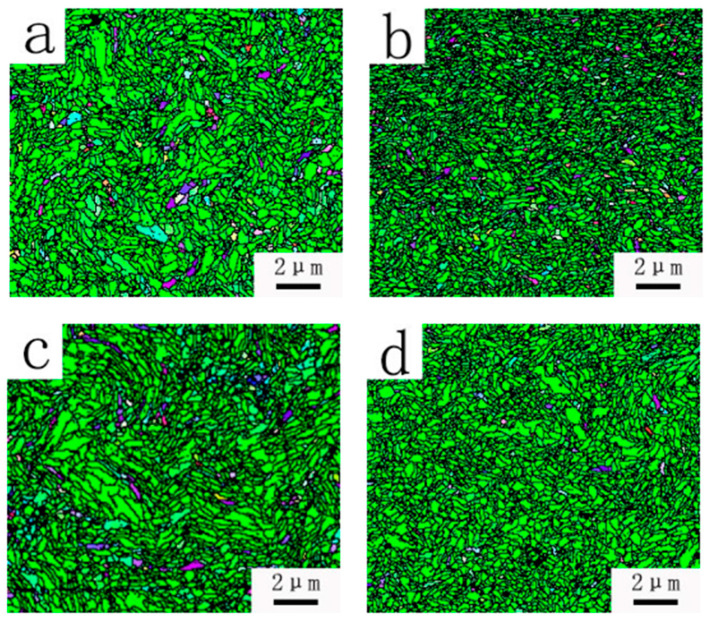
EBSD diagram of 0.50 mm tungsten alloy wire cross-section [83]: (**a**) process 1; (**b**) process 2; (**c**) process 3; (**d**) process 4.

**Figure 23 micromachines-14-01030-f023:**
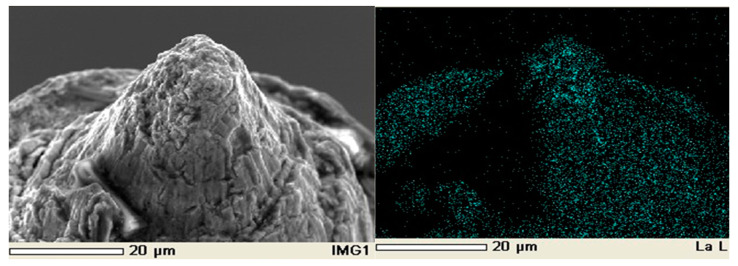
Fracture morphology of W–1.0% La_2_O_3_ alloy wire.

**Table 1 micromachines-14-01030-t001:** Particle size characteristics and pressed billet strength of tungsten powder prepared from different raw materials [21].

Raw Materials	Fsss Particle Size/μm	Span	Pressed Billet Strength/MPa
Purple tungsten	2.92	2.19	4.8
Blue tungsten	3.60	2.19	5.4
Yellow tungsten	3.16	1.65	4.4

**Table 2 micromachines-14-01030-t002:** Powder size of different manufacturing methods.

Manufacturing Method	Raw Materials	Powder Size	Advantages of Each Method
Plasma spheroidization method [22]	W	8–20 μm	Homogeneouspore distribution and an open pore channel
Gas atomization method [23]	Co, Cr, Mo, W	20–55 μm	High sphericity
Electric wire explosion method [24]	W	fine (100–300 nm)	Producing low-aggregated bimodal powders
		coarse (0.3–3.2 μm)	
Plasma Rotating Electrode Processing (PREP) [25]	W bar	15–53 μm	High sphericity, high purity and low impurity content
		45–150 μm	
Plasma Rotating Electrode Processing (PREP) [26]	W bar, La_2_O_3_	45–150 μm	High sphericity, high purity and low impurity content

**Table 3 micromachines-14-01030-t003:** Relative densities of different manufacturing methods.

Manufacturing Method	Materials	Relative Densities
Molding [27]	W	94.8%
Hot isostatic pressing [32]	90W–Ni–Fe–Cu	Above 98.5%
Hot isostatic pressing [33]	W/CuY	98.1%
	W/CuCrZr	98.8%
Hot isostatic pressing [34]	W	97.2%
Spark plasma sintering [34]	W	91.7%
Hot isostatic pressing) [35]	93W–Ni–Fe	98.8%

**Table 4 micromachines-14-01030-t004:** Texture volume fraction of pure tungsten plate prepared by different rolling methods [55].

Rolling Methods	{100}	{110}	{111}
UNR	29.62%	1.20%	14.61%
CRR	31.24%	7.23%	33.33%
CLR	24.90%	3.18%	33.92%

**Table 5 micromachines-14-01030-t005:** Elongation and tensile strength of sintered and swaged tungsten alloys [60].

Condition	Elongation (%)	Tensile Strength (MPa)
sintered	2.7 ± 0.6	615 ± 44
sintered and swaged	7.3 ± 0.4	1105 ± 13

**Table 6 micromachines-14-01030-t006:** Properties of tungsten alloys in sintered, rolled and annealed states [70].

States		Tensile Strength/MPa	Elongation/%
Sintered		850	15
Rolled		1215	3
Annealed	800	1215	3
	1000	1210	3
	1200	1050	8
	1450	900	14

**Table 7 micromachines-14-01030-t007:** Parameters of different drawing processes [83].

	Compression Rate (%)	Mold Temperature (°C)	Wire Heating Temperature (°C)
Process 1	22	550	1050
Process 2	15	550	950
Process 3	15	550	1050
Process 4	22	550	950

## Data Availability

Sorry, since the partner needs to keep the original data confidential, it cannot be shared.

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
