# Peer review of "Research Status of Manufacturing Technology of Tungsten Alloy Wire"

_micromachines, 2023, doi:10.3390/mi14051030_

Round 1
Reviewer 1 Report
This paper intends to provide a review on the research status of Tungsten alloy wire manufacturng technologies. However, this is merely a lengthy version of overview of how Tungsten alloy wires are made from scratch. This article is clearly out of focus.
1. Please proof read carefully. Grammar error on page 6 line 9.
2. While a number of published works are discussed for each manufacturing process, it is unclear what challenges are to be solved for further advancement of the fields. In fact, no useful conclusions can be drawn from this review other than that there are a lot of work done in this area. In order to qualify as a review paper, more meaningful conclusions need to be drawn and the amount of work discussed is clearly not enough considering the number of processes included. Each manufacturing method discussed can in fact be a topic of a review paper.
Readability is commendable. Proof read is still needed though.
Reviewer 2 Report
This article reviews the development of tungsten wire manufacturing technology. It includes powder manufacturing, pressing, sintering, rolling, rotary forging, annealing, drawing and other processes, wire drawing and other processes. This article is based on the analysis of the latest publications in this field.
In the reviewer opinion, the paper can be recommended for publication in micromachines journal after addressing the following comments:
- In the abstract, it is best to highlight the strengths of the article in relation to other review articles in the same field. Otherwise, authors should highlight the novelty of the current review article compared to what has already been done.
- The abstract and introduction are too short for a review article and should be expanded.
- Some references says that ``Global tungsten resources are mainly concentrated in the Alps Himalaya range and circum Pacific Geological belt. Russia's tungsten ore resources are concentrated in the North Caucasus, East Siberia, and the Far East of the tin Holt A Lin range, and the larger mine is Verkhne-Kayrakty``. Which is a little bit different with the first paragraph of the introductory section.
- In a review article, it is usually best to use tables that summarize the results of several research articles with reference to the analysis of a topic in the subject being reviewed. It is recommended for example in section 2 to draw a table that summarize how to prepare tungsten powder with references authors and finding.
- In Table 2, it is recommended that the benefits of each manufacturing method be added for comparison.
- It is recommended that a figure for each manufacturing process be added at the beginning of each section to facilitate understanding of the process.
- It is recommended that a table be added at the end of each section summarizing the results for each reference analyzed.
-Add a reference for the figure 8
- Add a reference for the figure 16, 17, 20.
- Copyright permission from the author is required for the various figures presented in the manuscript.
- There is no criticism given by the authors throughout the paper. It is recommended to add a discussion section.
- In a review article, it is recommended that authors develop their own tables and charts in which they make a comparison or analysis to make the critique of the references reviewed.
- In the discussion section, which should be added, the authors need to clarify for readers and new researchers the various strands and points that need further consideration in the future.
- The novelty of the review should be highlighted in the conclusion section.
- The references format should be revised and standardized
Round 2
Reviewer 1 Report
The new abstract is worse than the old abstract. The old abstract was ok except that it was too broad. Adding a few limiter adjectives to the old abstract would be sufficient, such as "brief", "in the context of", "tutorial", etc. The first few sentences of the new abstract should be in the introduction, not in the abstract.
N/A
Author Response
Thank you very much for your comments and suggestions. The abstract has been revised. Please see the article abstract for details.
Reviewer 2 Report
the updated version can be accepted
Author Response
Thank you very much for your comments and suggestions.The abstract has been slightly revised according to the suggestion of reviewer 1.Please see the article abstract for details.